# Heterozygous Deletion of Chromosome 15q13.3 in a Boy with Developmental Regression, Global Developmental Delay, Hypotonia, and Short Stature

Allison M. Strauss [1,*,†], Anna C. Buhle [1,†] and David M. Finkler [1,2]

1 Virginia Tech Carilion School of Medicine, Roanoke, VA 24016, USA
2 Department of Pediatrics, Carilion Clinic, Roanoke, VA 24014, USA
* Correspondence: ams2023@vt.edu
† These authors contributed equally to this work.

**Abstract:** Two causes of intellectual disability are 15q13.3 deletion syndrome and *BRWD3* X-linked intellectual disability. 15q13.3 deletion syndrome causes a heterogenous phenotype including intellectual disability (ID), developmental delay (DD), autism spectrum disorder, epilepsy/seizures, schizophrenia, attention deficit hyperactivity disorder, visual defects, hypotonia, and short stature. *BRWD3* variants are rare, and the clinical presentation is largely unknown. Presented here is a 34-month-old male with developmental regression, global DD, hypotonia, and short stature. In this study, the patient and his mother underwent a whole-genome array screening. Sorting intolerant from tolerant (SIFT) and polymorphism phenotyping v2 (PolyPhen-2) analyses were performed to determine the pathogenicity of the *BRWD3* mutation. Array comparative genomic hybridization showed a heterozygous, pathogenic deletion of at least 1.6 Mb from the cytogenetic band 15q13.2q13.3 and a *BRWD3* variant of unknown clinical significance. This combination of genetic mutations has never been reported together and neither disorder is known to cause developmental regression. The mechanism of developmental regression is undefined but is of great importance due to the opportunity to develop therapies for these patients.

**Keywords:** 15q13.3 heterozygous deletion; microdeletion; BRWD3; developmental regression



## 1. Introduction

There are many genetic disorders that are known to cause intellectual disability (ID) in patients, including 15q13.3 deletion syndrome. Sharp et al. first described the 15q13.3 microdeletion by using microarray-based comparative genomic hybridization to search for genetic causes of ID in a large number of patients [1]. Since then, many individuals have been discovered with a 15q13.3 deletion. This chromosomal change in the long arm of chromosome 15 is linked to an array of highly variable clinical presentations, including ID, epilepsy, abnormal electroencephalogram (EEG) findings, and mild, variable facial dysmorphism [1]. Others include muscular hypotonia, visual dysfunction, schizophrenia, and mood disorders [2,3]. Major congenital malformations are uncommon and some patients may not experience any intellectual, behavioral, or physical features at all [3]. When the mode of inheritance is known, 15q13.3 microdeletions are primarily maternal in origin, however de novo deletions may occur [3,4]. Most are heterozygous with high variability in the expression and penetrance of clinical symptoms. This is in contrast to the less common homozygous microdeletions that tend to express highly consistent features including encephalopathy, hypotonia, developmental delay (DD), cortical vision impairment, epilepsy, and abnormal EEG findings [5]. DD is defined as failure to achieve developmental milestones (including gross motor, fine motor, language, and social) compared to peers of the same age range [6]. Developmental regression is defined as loss of previously acquired developmental milestones [6].

The BRWD3 gene is believed to be involved in the development and regulation of neural connections [7]. BRWD3 maps to Xq21.1 and encodes for a bromodomain and WD-repeat domain-containing protein [8]. Patients with pathologic BRWD3 mutations present with ID, motor delay, speech difficulty, tall stature, obesity, facial dysmorphia, macrocephaly, and behavioral disturbances, as well as many other phenotypes [9]. BRWD3 variants are rare, and the clinical presentation is not well understood [9–13].

Reported here is a patient with both 15q13.3 deletion syndrome and a *BRWD3* variant with DD, developmental regression, short stature, and global hypotonia. This is the first case describing regression in a patient with 15q13.3 deletion syndrome.

## 2. Case Presentation and Examination

The patient is a 34-month-old male with an 82.6 cm length (<1st percentile) and 49.5 cm head circumference (50th percentile). He was born at 40 weeks 6 days gestational age via spontaneous vaginal delivery. APGAR scores were 8 and 9 at 1 and 5 min, respectively. Pregnancy was complicated by chorioamnionitis and maternal buprenorphine use. His first year of life was uncomplicated. Later, he had gross motor delays including walking at 24 months (typically occurs at 12–15 months) and speaking 20 words at 32 months (typically occurs by 18 months). At 34 months, the patient was admitted to pediatric care for the chief complaint of failure to thrive and developmental milestone regression. Physical examination showed a 10.3 kg weight (<1st percentile), 0.825 m height (5th percentile), normocephaly (50th percentile), with no apparent deformities. His weight was decreased (50th percentile to <1st percentile) with a significant deceleration in his height velocity as well. There was initial concern for an endocrine abnormality; however, comprehensive metabolic panel, adrenocorticotropic hormone, and insulin-like growth factor 1 were normal. CBC showed mild anemia. A CT scan showed gastroparesis and megalogastria which were thought to be contributing to the failure to thrive, at least in part.

Upon admission, the patient was practically non-verbal but alert and playful. He said "mama" and pointed to get attention, walked with an unsteady wide-based gait, and had normal fine motor skills. The patient's mother reports that 2 months prior, he was able to say 20 words and walked with a normal gait without cruising. His current presentation is a significant regression from his mother's reporting. Neurologic exam showed global hypotonia, limb wasting, and mild weakness. Muscle stretch reflexes appeared normal. Eye examination showed isocoric pupils, normal visual tracking, and eye contact with other people. A work-up for his gross DD and developmental regression included EEG, brain magnetic resonance imaging (MRI), and creatine kinase lab, all of which were normal. An autism/ID panel was performed and detected a 15q13.3 microdeletion of at least 1.6 Mb and a *BRWD3* variant.

The mother is in good health; however, she has a history of supraventricular tachycardia corrected with ablation and mitral valve prolapse. The father's health history is not known. The patient is the only child of both parents and there is no known consanguinity.

## 3. Diagnostic Investigation

A whole-genome array of genomic DNA was performed on the patient and his mother (GeneDx, Gaithersburg, MD, USA) using a proprietary capture system developed by GeneDx for next generation sequencing with CNV calling (NGS-CNV). Bi-directional sequence reads were assembled and aligned to reference sequences based on NCBI RefSeq transcripts and human genome build GRCh37/UCSC hg19. Genomic imbalances are reported using UCSC human genome build 19 (NCBI build 37, February 2009).

## 4. Outcome

Array comparative genomic hybridization showed a heterozygous, pathogenic deletion of at least 1.6 Mb from the cytogenetic band 15q13.2q13.3(*30,954,726-32,509,926)x1*. The deletion contained eight genes: *MMTR10, TRPM1, MIR211, KFL13, OTUD7A, CHRNA7, GOLGA8K,* and *GOLGA8O.* These results are consistent with the diagnosis of 15q13.3

deletion syndrome. Parental testing of the mother was done and showed that she does not harbor the 15q13.2q13.3 deletion. The father was not available for genetic testing. Additionally, the patient is hemizygous for a variant of the X-linked *BRWD3* gene. This variant, p.Ile705Thr (ATT > ACT) c.2114 T > C in exon 19, was inherited from his mother.

## 5. Discussion

The variability of clinical presentations is vast for patients with heterozygous 15q13.3 microdeletion [1–3]. The patient's 1.6 Mb microdeletion comprised the break point (BP) BP4–BP5–BP6 region. BP4–BP5 is the most common region impacted in 15q13.3 microdeletion, with a more distal microdeletion comprising BP5–BP6 reported rarely in the literature [1,14].

Of the eight genes deleted in the patient, three key genes have been associated with phenotypic characteristics: *CHRNA7*, *TRPM1*, and *OTUD7A* [15,16]. *CHRNA7* encodes ligand-gated ion channels that mediate fast signal transmission at synapses and is highly prevalent in the brain [15]. Patients with *CHRNA7* deletions have been identified as having high susceptibility for epilepsy [17–19]. Additionally, it is thought that *CHRNA7* deletions may contribute to the cognitive defects sometimes seen in 15q13.3 deletion syndrome [15]. The *TRPM1* gene is thought to play a role in the ophthalmic findings associated with 15q13.3 microdeletion [20]. The *OTUD7A* gene encodes a deubiquitinating enzyme expressed in the brain that may be involved in neurodevelopment [16,21]. The phenotype of the patient is mostly consistent with that of 15q13.3 deletion syndrome. ID is common; in a case series of 246 patients with 15q13.3 microdeletion, 80% had at least one neuropsychiatric diagnosis, with DD or ID accounting for the largest percent (57.7%) [3,16]. This large case series, as well as other case series and reports, note similar findings of ID in patients with 15q13.3 microdeletion [3]. The patient presented with a global DD, hypotonia (present in 13.5% of cases), short stature (present in 26% of cases), and failure to thrive, all of which are consistent with 15q13.3 microdeletion syndrome [3]. He did not show signs of epileptic activity or seizures (present in 36.8% of cases), dysmorphic facial features (present in 13.6% of cases), or visual impairment (present in 5.9% of cases) [3,22].

The patient is additionally harboring a hemizygous *BRWD3* variant which was inherited from his mother. This variant is unique and has not been reported in the literature to the best of our knowledge. Polymorphism Phenotyping v2 (PolyPhen-2) analysis determined that this missense mutation is possibly damaging to the function of the gene. Sorting intolerant from tolerant (SIFT) analysis determined that the substitution is likely to be tolerated. It is unknown if this was truly a pathogenic mutation or if the patient's clinical presentation was due solely to the 15q13.3 deletion. These common clinical characteristics have been summarized in the Table 1 below for comparison.

**Table 1.** Summary of common clinical features in 15q13.3 microdeletion syndrome and BRWD3 variants compared to the presenting patient.

| 15q13.3 Microdeletion Syndrome | BRWD3 Variants | Presenting Patient |
|---|---|---|
| DD/ID | ID | Developmental regression |
| Hypotonia | Motor delay | Global DD |
| Short stature | Speech difficulties | Hypotonia |
| Epileptic activity/seizures | Tall stature | Short stature |
| Dysmorphic facial features | Obesity | Failure to thrive |
| Visual impairment | Facial dysmorphia | |
| Schizophrenia | Macrocephaly | |
| Mood disorders | Behavioral disturbances | |

The developmental regression was notable in this case for being unique in patients with 15q13.3 microdeletion syndrome. To the best of our knowledge, this is the first case of a child with a 15q13.3 microdeletion presenting with a loss of developmental milestones

previously reached. The pathogenic mechanisms underlying developmental regression are poorly understood and are a topic of great interest. Understanding the underlying cause of developmental regression could lead to individually tailored medical care for patients with regressive phenotypes. There have been several proposed biological mechanisms of developmental regression, including the synaptic "over pruning" hypothesis in which the normal breakdown of synaptic connections that occurs during neurodevelopment occurs in excess [23]. Both *CHRNA7* and *OTUD7A* are thought to be involved in the development of the nervous system. It is feasible that a deletion of either of these genes may contribute to the regression that was seen in this patient. Alternatively, it may be that the combination of both the 15q13.3 deletion and the *BRWD3* mutation caused developmental regression in the patient. However, it is uncertain if this *BRWD3* variant is pathogenic, so it cannot be determined if it contributed to the patient's phenotype.

## 6. Conclusions

Reported in this case is a 34-month-old male with both 15q13.3 deletion syndrome and a hemizygous *BRWD3* variant. While his clinical presentation includes some features common to these diagnoses, his presentation of developmental regression is distinct. With the mechanism of developmental regression still undefined, this unique case adds to the literature data that could be used to further elucidate the pathogenesis. Additionally, genetic disorders such as 15q13.3 deletion syndrome should be on the differential diagnosis for a patient presenting with developmental regression.

**Author Contributions:** A.M.S. and A.C.B. aided with case report conceptualization and design, literature review and investigation, and were manuscript authors and editors. D.M.F. provided supervision and oversight, and was a manuscript author and editor. All authors have read and agreed to the published version of the manuscript.

**Funding:** The authors received no financial support for the research, authorship, and/or publication of this article.

**Institutional Review Board Statement:** Ethical review and approval were waived for this study, as it was not applicable for a case report describing a single patient.

**Informed Consent Statement:** Informed consent was obtained from all subjects involved in the study.

**Data Availability Statement:** No datasets were generated or analyzed during the current study.

**Acknowledgments:** The authors thank Emily Doherty, M.D. We also wish to acknowledge the patient and his family for participating in this case report.

**Conflicts of Interest:** The authors declare no conflict of interest.

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
