# Peer review of "Heterozygous Deletion of Chromosome 15q13.3 in a Boy with Developmental Regression, Global Developmental Delay, Hypotonia, and Short Stature"

_pediatrrep, doi:10.3390/pediatric14040061_

Round 1

Reviewer 1 Report

The aim of Heterozygous deletion of chromosome 15q13.3 in a boy with developmental regression, global developmental delay, hypotonia, and short stature was to acknowledge a novel combination of a 15q13.3 deletion and BRWD3 variant as a potential influence on the developmental regression in a young, male patient. This case study was well written and should be accepted after minor revisions. Additionally, we are happy to review the resubmission.

Abstract

  • In line 21, the word “Two” should not be bolded.
  • On lines 27 and 28, the “SIFT” and “PolyPhen-2” abbreviations should be defined.

Introduction

  • Like in the abstract, abbreviations should again be defined. Examples are “ID” in line 37, “PolyPhen-2” in line 126, and “SIFT” in line 127.”
  • More background information is needed for the BRWD3 gene.
  • Developmental delay and regression should be defined (i.e. does this involve speech, motor, cognitive, what other skills were lost, etc.?)

Case Presentation and Examination

  • Was there more information regarding the patient’s developmental regression besides his mother’s reporting? Had he been seen over time by specialists who also noted the extinction of these behaviors? 
  • Based on the definition of developmental delay and regression, what else did the patient show that would allow for this to be the diagnosis?

Discussion

  • In line 117, the statement “The large case series…15q13.3 microdeletion” should have an accompanying reference.
  • In line 119, the statement “The patient presented with…microdeletion syndrome” should have an accompanying citation regarding how the stated presentations were associated with the deletion.
  • In line 129, the statement “BRWD3 maps to…domain-containing protein” should have an accompanying reference.

Reviewer 2 Report

The manuscript by Allison M. Strauss, Anna C. Buhle and David M. Finkler is clearly presented and contributes to enriching the knowledge of rare pediatric diseases. I report below few suggestions to the Authors:

-In the section "Introduction, I would write the acronym of ID in full, even if it has already been abbreviated in the abstract.

-Most of the sentences are very short, I would suggest unifying some sentences that describe the same topic (for example line 40-44).

-The section "Conclusions" could be enriched by giving greater emphasis to the importance of the examined clinical case.

-The manuscript contains few bibliographic notes, if possible, I would suggest the Authors to enrich the references.

-I would suggest the authors to insert a summary table of the clinical feature of the patient.
